# CLOSE THE GAP: LIGHTWEIGHT IMAGE CAPTIONING VIA RETRIEVAL AUGMENTATION

## ABSTRACT

Image Captioning is important for many applications such as content-based image search or accessibility for visually impaired individuals. To achieve rich language capabilities, recent work conditioned pretrained language models (LMs) on pretrained vision-language models (VLMs) that allow for image inputs. However, pretrained VLMs usually suffer from a modality gap which constitutes the misalignment of image and text representations in the joint embedding space. While this gap can in principle be minimized by finetuning, this is usually costly or often infeasible and requires large amounts of task specific data. To address this issue, we propose to bridge the modality gap at lower costs via a linear mapping that is optimized via a least-squares solution. This does not require gradients and can be computed within minutes, even on CPU. At inference, we apply our mapping to images embedded by the VLM and retrieve the closest captions from the training set. Along with an instruction, these captions serve as a prompt for the LM to generate a new caption. In addition, we propose a method to iteratively refine the mapping by bootstrapping synthetic captions from the LM. This enables explicit optimization for commonly used image captioning metrics. We find that a reference-free metric, namely the CLIP-score, often assign high scores to hallucinated content. On reference-based metrics, our method achieves competitive performance to lightweight captioning approaches on MS-COCO and Flickr30k datasets.[1]

## 1 INTRODUCTION

The task of image captioning aims at understanding the relationship between visual and textual data and requires generative capabilities on the textual side. It requires machines to generate informative descriptions for images, which can be useful in various applications such as image retrieval, content-based image search, and accessibility for visually impaired individuals (Gurari et al., 2020).

Recent works have advanced the state of the art in image captioning by leveraging off-the-shelf foundation models (FMs, Bommasani et al., 2021) combined with large-scale pretraining and finetuning for transfer to new domains. However, this paradigm induces substantial computational cost. A recent trend referred to as *lightweight captioning* aims at reducing the computational cost by updating only a small amount of parameters during training. Using pretrained VLMs enables retrieval augmentation for lightweight image captioning, which has proven to be effective in practice (Ramos et al., 2022; 2023). These works rely on a pre-trained CLIP model (Radford et al., 2021) to retrieve captions from a datastore that are similar to an input image.

However, CLIP suffers from the so-called modality gap (Liang et al., 2022), which refers to a mis-alignment of images and texts in the joint embedding space. To bridge this gap, prior image captioning pipelines require training in an end-to-end manner (Ramos et al., 2022; Luo et al., 2023; Mokady et al., 2021). Our aim is to bridge the modality gap prevalent in CLIP with a simple and lightweight mapping that does not require end-to-end training. This allows us to close the modality gap for the downstream task of image captioning while retaining competitive performance compared to other lightweight captioning approaches.

We propose to mitigate the modality gap via a linear mapping to enable lightweight image captioning with retrieval augmentation. First, we compile a dataset of image and text correspondences in the

---

[1]We will make our code publicly available upon publication.

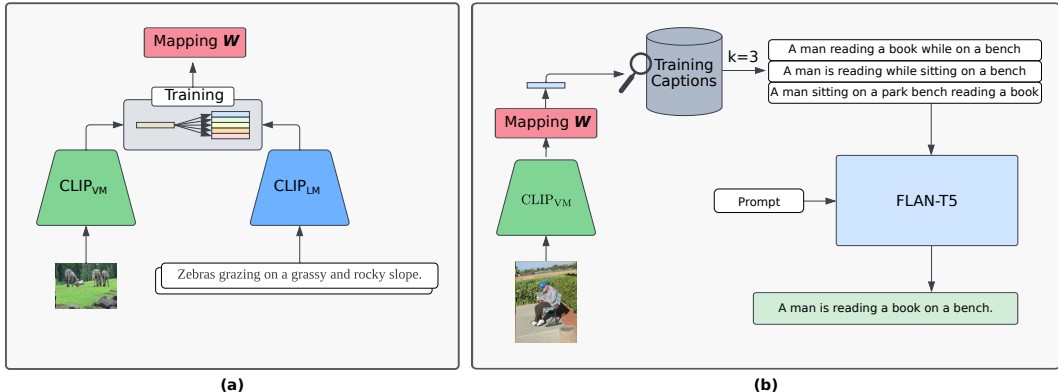

Figure 1: **(a)** We train a linear mapping $W$ to bridge the modality gap prevalent in CLIP. **(b)** On inference, we employ the computed mapping to retrieve captions that are similar to the input image and provide these along with a prompt to a FLAN-T5 LM.

joint embedding space of CLIP by using a publicly available dataset. Then, we compute our mapping via a least-squares solution in closed form on CPU. At test time, we embed an unseen image in the CLIP embedding space and apply our mapping. Next, we retrieve the closest texts to an image and feed them along with a prompt to a generative LM to generate a new caption. Moreover, we introduce a self-improvement loop that iteratively augments the training set for our mapping with synthetic captions generated by our base method. To obtain a set of synthetic captions for a training image, we sample a set of new captions from the LM. Then, we only add synthetic captions to our training set that achieve a high score according to a commonly used captioning metric. This allows us to indirectly optimize the mapping toward a certain metric. Our method provides a cheap and efficient manner to enhance retrieval augmented captioning methods that use a pretrained VLM by mitigating the modality gap.

We evaluate our lightweight image captioning pipeline on two popular benchmarks, namely MS-COCO (Lin et al., 2014) and Flickr30k (Young et al., 2014). Our method achieves competitive performance on both datasets, with only 1 M trainable parameters. Moreover, we investigate transfer capabilities of our method across domains (e.g., MS-COCO to Flickr30k) where our method outperforms other lightweight retrieval-based approaches. The self-improvement loop slightly increases the performance providing further evidence that synthetic captions can improve captioning performance. We observe that when optimizing for the commonly used reference-free CLIP score, our method tends to generate hallucinated content. Contrary, when we filter according to rule-based evaluation metrics, we observe improvements coherent across different metrics. We further investigate this phenomenon and demonstrate that CLIP score generally assigns high scores to hallucinated content or even a simple bag of words. Our contributions are as follows:

- We introduce a novel method for lightweight image captioning with only 1 M trainable parameters that reaches competitive performance to prior lightweight captioning approaches
- We show that synthetic captions bootstrapped by pretrained LMs can be used to further improve our method on the downstream task of image captioning
- We demonstrate that CLIP-score (Hessel et al., 2021), a recently proposed reference-free evaluation metric, is vulnerable to hallucinated content and bag of words

## 2 METHODS

We propose a novel method for retrieval-augmented image captioning, which we call ReCap. ReCap leverages pretrained VLMs to retrieve captions that are similar to a given image. Then we feed these captions to a pretrained LM along with a prompt to generate a new caption. However, the retrieval is affected by the well-known modality gap present in pretrained VLMs (Liang et al., 2022). ReCap aims to mitigate this gap via a lightweight linear mapping which can be computed via a closed-form solution. The key is that we can optimize that mapping for the task/dataset at hand, thus sidestepping the need for end-to-end finetuning.

## 2.1 Closing the Gap

We assume access to a dataset $\mathcal{D} = \{(\boldsymbol{x}_i, \boldsymbol{c}_i)\}$ that provides image-text pairs, e.g., MS-COCO (Lin et al., 2014). First, we embed the images of the training split $\mathcal{D}_{\text{Train}} \subset \mathcal{D}$ using a CLIP vision encoder $\phi : \mathcal{X} \to \mathbb{R}^d$, where $\mathcal{X}$ is the pixel space and $d$ denotes the dimension of embedding space. This results in an image embedding matrix $\boldsymbol{F}_{\mathcal{D}_{\text{Train}}} = (\boldsymbol{f}_1, \ldots, \boldsymbol{f}_n)^\top \in \mathbb{R}^{n \times d}$. Then we embed the corresponding captions via the CLIP text encoder $\psi : \mathcal{C} \to \mathbb{R}^d$ to obtain $\boldsymbol{E}_{\mathcal{D}_{\text{Train}}} = (\boldsymbol{e}_1, \ldots, \boldsymbol{e}_n)^\top \in \mathbb{R}^{n \times d}$, where $n$ denotes the number of embedded captions in $\mathcal{D}_{\text{train}}$. If, like in the case of MS-COCO, we are presented with multiple captions per image, then we assume the same image just appears multiple times in $\mathcal{D}$ (see Figure 1, (a)). Finally, we fit a least-squares linear model $\boldsymbol{W} \in \mathbb{R}^{d \times d}$ with inputs $\boldsymbol{F}_{\mathcal{D}_{\text{Train}}}$ and targets $\boldsymbol{E}_{\mathcal{D}_{\text{Train}}}$ so that $\|\boldsymbol{W} \boldsymbol{f}_i - \boldsymbol{e}_i\|^2$ becomes minimal for all $(\boldsymbol{x}_i, \boldsymbol{c}_i) \in \mathcal{D}_{\text{train}}$. The linear model $\boldsymbol{W}$ bridges the modality gap between image and text modalitites. The solution to the least-squares problem has a time complexity of $\mathcal{O}(d^3)$.

## 2.2 Image Captioning

Using our semantic mapping $\boldsymbol{W}$ we can pair a vision encoder with a generative LM to facilitate language generation conditioned on visual input (see Figure 1). Given an image $\boldsymbol{x} \in \mathcal{X}$, we compute an embedding $\boldsymbol{f} = \phi(\boldsymbol{x})$ and select the set $\mathcal{E}$ of top-$k$ targets by

$$\mathcal{E} = \arg \max_{i \in \{1, \ldots, n\}}^{k} \text{cossim}(\boldsymbol{e}_i, \boldsymbol{W} \boldsymbol{f}), \tag{1}$$

where $\arg \max^k$ denotes an extension of the $\arg \max$ operator returning the arguments of the $k$ largest elements of a set and

$$\text{cossim}(\boldsymbol{a}, \boldsymbol{b}) = \frac{\boldsymbol{a}^\top \boldsymbol{b}}{\|\boldsymbol{a}\| \|\boldsymbol{b}\|} \tag{2}$$

is the cosine similarity. The retrieval process has a complexity of $\mathcal{O}(n)$, where $n$ is the number of elements to retrieve from. The retrieved targets in $\mathcal{E}$ are provided to a generative LM as context along with a prompt to generate a new caption for the image $\boldsymbol{x}$. Algorithm 1 describes the procedure on how we perform image captioning via ReCap.

## 2.3 Iterative Self-Improvement

We can refine $\boldsymbol{W}$ by augmenting $\mathcal{D}_{\text{train}}$ with synthetic captions for images in the training set. Our aim is to only add synthetic captions of high quality to $\mathcal{D}_{\text{train}}$ so that the over-all prediction quality of our model improves. To this end, we assume access to an image captioning metric $m(\cdot, \cdot)$ that takes a candidate and a set of reference captions as input and returns a scalar value. Then, we evaluate ReCap on the validation set and compute the average metric $\bar{m}$, which provides us with an estimate of the quality of generated captions. Next, we generate a set of new captions for images in $\mathcal{D}_{\text{train}}$ by sampling from the LM. We compute $m(\cdot, \cdot)$ for each synthetic caption and only keep those for which their score exceeds $\bar{m}$. After generating synthetic captions for all images in $\mathcal{D}_{\text{train}}$, we add them to our training set and our datastore and re-train $\boldsymbol{W}$. Then we again evaluate performance on the validation set for the new $\boldsymbol{W}$ and update $\bar{m}$. We repeat this process for several rounds until we do not observe any improvement in $\bar{m}$ anymore. Algorithm 2 shows the pseudocode for our proposed self-improvement loop.

---

**Algorithm 1** Image captioning via ReCap

**Require:** CLIP vision encoder $\phi(\cdot)$, CLIP text encoder $\psi(\cdot)$, Training set $\mathcal{D}_{\text{Train}} = \{(\boldsymbol{x}_i, \boldsymbol{c}_i)\}$, Test set $\mathcal{D}_{\text{Test}} = \{(\boldsymbol{x}_j)\}$, Hyperparameter $k$, Language Model $\text{LM}(\cdot)$, Prompt $\mathcal{P}$

$\{(\boldsymbol{f}_i, \boldsymbol{e}_i)\}_{i=1}^{|\mathcal{D}_{\text{Train}}|} \leftarrow \phi(\boldsymbol{x}_i), \psi(\boldsymbol{c}_i)$ for $(\boldsymbol{x}_i, \boldsymbol{c}_i) \in \mathcal{D}_{\text{Train}}$      ▷ Embed training set
$\boldsymbol{W} \leftarrow \texttt{fit\_linear}(\{(\boldsymbol{f}_i, \boldsymbol{e}_i)\})$      ▷ Pre-compute linear mapping
$\mathcal{B} \leftarrow \{\boldsymbol{e}_i\}$      ▷ Initialize datastore with training captions
$\{\boldsymbol{W} \boldsymbol{f}_j\}_{j=1}^{|\mathcal{D}_{\text{Test}}|} \leftarrow \phi(\boldsymbol{x}_j)$ for $(\boldsymbol{x}_j, \boldsymbol{c}_j) \in \mathcal{D}_{\text{Test}}$      ▷ Embed test images in CLIP space
$\{\mathcal{E}_j\} \leftarrow \texttt{topk}(\{\boldsymbol{W} \boldsymbol{f}_j\}, \mathcal{B}, k)$      ▷ Retrieve top-k captions from datastore
$\{\mathcal{S}_j\} \leftarrow \text{LM}(\texttt{concat}(\mathcal{P} + \mathcal{E}_j))$      ▷ Generate new captions

## 3 EXPERIMENTS

In this section, we first describe the experimental setup in Section 3.1. Then we present results for image captioning on the established benchmarks MS-COCO (Lin et al., 2014) and Flickr30k (Young et al., 2014) in Section 3.2. Further, we assess the cross-domain transfer capabilities of our method from MS-COCO to Flickr30k in Section 3.3. We present ablation studies on our mapping in Section 3.4 and find the best form of language supervision, ranging from single-token level to narrative level. Finally, Section 3.5 shows qualitative results for our retrieval, correlations between commonly used metrics, and CLIP-score's vulnerability to hallucinations.

### 3.1 EXPERIMENTAL SETUP

We split both benchmark datasets according to Karpathy & Fei-Fei (2017) into training, validation, and test splits. As preprocessing we perform length normalization and mean centering of both image and caption embedding vectors as suggested by (Artetxe et al., 2016). We found mean centering of the embedding spaces to be extremely important. Then we compute our mapping on image-caption pairs of the respective train split via ordinary least squares. Importantly, the number of parameters for our mapping varies with the dimensionality $d$, which is at most 1024. To find the best setting for image captioning, we search over different vision encoders, LMs, decoding strategies, and prompt ordering. Moreover, we search over multiple values of retrieved texts ($k$). For more details about hyperparameter search and choice of encoders or decoders, see Appendix A. We use faiss (Johnson et al., 2019) to manage our datastore, since it enables efficient storage and retrieval from vector databases. Our final setting uses a RN50x64 CLIP encoder[2] and a FLAN-T5-Large (Chung et al., 2022). All generative LMs used in our work are publicly available on the huggingface hub (Wolf et al., 2020). To generate captions with FLAN-T5, we use the same prompting strategy as used in (Ramos et al., 2022). Specifically, the used prompt template is "Similar images show: {} This image shows: ", where the most similar captions are inserted instead of the curly brackets. We experimented with different prompts, such as summarization, which lead to slightly worse results. Regarding the self-improvement loop we experimented with different metrics to threshold the quality of synthetic captions. We found that CIDEr-D (Vedantam et al., 2015) is well suited and usually leads to a slight improvement for all other metrics as well.

We report metrics commonly used for image captioning, such as BLEU-4 (B@4, Papineni et al., 2002), ROUGE-L (R-L, Lin & Och, 2004), CIDEr-D (Vedantam et al., 2015), and SPICE Anderson et al., 2016[3]. Most prior works do not report error bars on metrics used for evaluation. We consider error bars to be very important as they indicate the variability of the measurements. Therefore, we provide them for all our evaluations in the form of the standard error.

### 3.2 BENCHMARK RESULTS

**MS-COCO**   We show results for ReCap on MS-COCO in Table 1. ReCap carries the least amount of trainable parameters (1 million) and is by far superior to competitors in terms of training time. Even though ReCap uses a substantially lower training budget, it reaches performance close to SmallCap in terms of SPICE. Considering n-gram based metrics, there is still a considerable gap between them. Inference time is approximately equal for SmallCap and ReCap (approximately 0.5 seconds on average on a TITAN-V). Using our self-improvement loop (ReCap+SelfImprove) we can improve upon SmallCap in some metrics, e.g. SPICE, where we observe a significant improvement after two iterations. This effect was not prevalent for the other metrics, where we observed a slight decrease after one iteration on the test set. We show captions generated via ReCap and SmallCap for randomly sampled images of the MS-COCO test set in Figure 2.

**Flickr30k**   We compare ReCap and ReCap+SelfImprove to I-Tuning and ClipCap, since these are the only other lightweight captioning methods that reported results on Flick30k. The results are shown in Table 2. ReCap achieves a slightly lower score in terms of CIDEr-D and SPICE. However, ReCap+SelfImprove is capable of closing this gap entirely after only one iteration of

---

[2]Taken from the official repository at https://github.com/openai/CLIP
[3]We evaluate our methods using the code from https://github.com/tylin/coco-caption

Table 1: Comparison of different lightweight methods on the MS-COCO test set. We show performance for ReCap with and without our self-improvement loop. We report mean and standard error for our methods. Results for other methods are taken from their respective publications. n/a indicates that a certain metric is not available for a given method. * indicates that self-improvement loop was performed for each metric separately. † indicates that training time must be multiplied by number of self-improvement iterations.

| | BLEU@4 | CIDEr-D | SPICE | $|\theta|$ | Training |
|---|---|---|---|---|---|
| CaMEL (Barraco et al., 2022) | 39.1 | 125.7 | 22.2 | 76 | n/a |
| ClipCap (Mokady et al., 2021) | 33.5 | 113.1 | 21.1 | 43 | 6h (GTX1080) |
| I-Tuning$_{Base}$ (Luo et al., 2023) | 25.2 | 116.7 | 16.9 | 14 | n/a |
| LLama-Adapter$_{V2}$ (Gao et al., 2023) | 36.2 | 122.2 | n/a | 14 | n/a |
| SmallCap$_{d=4,Base}$ (Ramos et al., 2022) | 36.0 | 117.4 | 21.0 | 1.8 | 8h(A100) |
| ReCap | $31.0 \pm 0.4$ | $107.4 \pm 1.0$ | $20.8 \pm 0.1$ | 1.0 | $20.3 \pm 1.91$s (CPU) |
| ReCap + SelfImprove* | $28.2 \pm 0.3$ | $103.0 \pm 0.9$ | $21.2 \pm 0.1$ | 1.0 | $20.3 \pm 1.91$s (CPU)$^{†}$ |

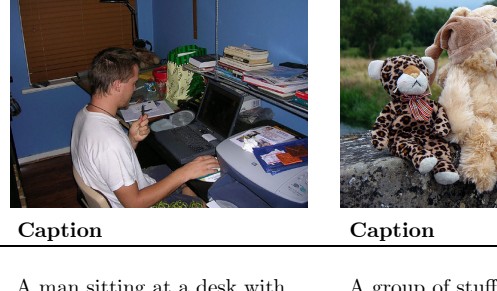
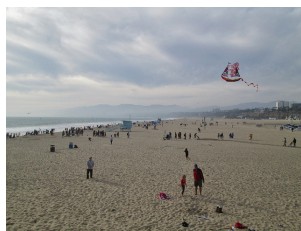

| Method | Caption | Caption | Caption |
|---|---|---|---|
| ReCap | A man sitting at a desk with a laptop computer on it. | A group of stuffed bears are stting down together. | A group of people flying kits on a beach |
| SmallCap | a man sitting at a desk with a laptop. | a group of teddy bears sitting next to each other. | a group of people standing on top of a beach. |

Figure 2: Captions generated via ReCap and SmallCap for four randomly sampeld images of the MS-COCO validation set.

self-improvement. Beyond that we did not observe any further improvements. Contrary, we observe a slight decrease for SPICE after one iteration of self-improvement.

## 3.3 CROSS-DOMAIN TRANSFER

Next, we investigate the cross-domain transfer of ReCap from MS-COCO to Flickr30k. We show results for two settings, (i) transferring the mapping while using in-domain data, and (ii), transferring both the datastore and the mapping. Further we show results for using an orthogonality constraint on the mapping (ReCap$_{Pr}$), since this has shown to be effective for closing the modality gap in prior work (Ouali et al., 2023). Table 3 summarizes the results. We only compare to SmallCap since it is the only other lightweight captioning method that uses retrieval augmentation. ReCap$_{Pr}$ attains the highest CIDEr-D score, significantly improving upon SmallCap, while ReCap exhibits only a slight improvement upon SmallCap. Interestingly, this

Table 3: Cross-domain transfer from MS-COCO to Flickr30k for SmallCap and ReCap. We report mean and standard error for ReCap. Results for other methods are taken from their respective publications.

| Method | CIDEr-D |
|---|---|
| **In-domain Datastore** | |
| SmallCap (Ramos et al., 2022) | 55.4 |
| ReCap$_{OLS}$ | $56.2 \pm 1.8$ |
| ReCap$_{Pr}$ | $\mathbf{59.7 \pm 1.9}$ |
| **OOD Datastore** | |
| SmallCap (Ramos et al., 2022) | $\mathbf{52.2}$ |
| ReCap$_{OLS}$ | $42.9 \pm 1.5$ |
| ReCap$_{Pr}$ | $44.2 \pm 1.6$ |

Table 2: Benchmark results for image captioning on the Flickr30k benchmark. We report mean and standard error for ReCap and ReCap+SelfImprove along with the number of trainable parameters $|\theta|$. Results for other methods are taken from their respective publications. * indicates that self-improvement loop was performed for each metric separately.

| Method | CIDEr-D | SPICE | $|\theta|$ |
|---|---|---|---|
| ClipCap (Mokady et al., 2021) | 57.9 | 15.8 | 43 |
| I-Tuning$_{\text{Base}}$ (Luo et al., 2023) | 61.5 | 16.9 | 14 |
| ReCap | $64.4 \pm 2.0$ | $15.9 \pm 0.3$ | **1** |
| ReCap + SelfImprove* | $\mathbf{66.4 \pm 1.9}$ | $\mathbf{17.1 \pm 0.3}$ | **1** |

only concerns the case for transfer of the mapping to data it was not trained on. This indicates that ReCap is more effective in leveraging new data in a training-free manner.

## 3.4 ABLATION STUDIES

We illustrated that ReCap is competitive with other lightweight captioning approaches, while requiring substantially less compute during training. Next, we perform an ablation study to assess the importance of the linear mapping to bridge the modality gap. We provide qualitative examples for retrievals with and without our mapping in Figure 5. Without the mapping, CLIP retrieves captions that describe semantically related contents to an image, which might not always be depicted in the image. Our mapping corrects for that and aligns the images with captions that describe contents present in the human annotated captions.

Further, we also consider different levels of language abstractions as target vectors $\boldsymbol{e}_i$ for computing the mapping. Specifically, we consider single tokens, prompt-augmented tokens [4], single captions, multiple captions (AllCaps), and finally, narratives. We obtain the token-level abstraction by tokenizing the training captions and using these as targets.[5] For AllCaps we concatenate all captions for an image into a single string and use the resulting embedding as target for an image. For narratives we take captions provided by the localized-narratives dataset (LN, Pont-Tuset et al., 2020). Depending on the level of abstraction we also change the datastore we retrieve from. That is, if we train the mapping on narratives, every entry in the datastore represents a narrative of the training set. Importantly, the different forms of language supervision result in different optimization problems. For token and caption level we have one-to-many relationships between input image and targets, while for AllCaps and Narratives we have one-to-one relationships. We assess all setups with and without linear mapping. We refer to methods that do not utilize the linear mapping as ReCap$^-$.

Additionally to CIDEr-D and SPICE, we report the recently proposed CLIP-score (CLIP-S), and RefCLIP-score (CLIP-RS) (Hessel et al., 2021) in Table 4. CLIP-S is a reference-free metric based on the scaled cosine similarity of image and the candidate caption in the joint embedding space of CLIP. CLIP-RS forms a harmonic mean between CLIP-S and the maximum cosine similarity of the image to reference captions, thus it is reference-based. As expected, we observe a drastic drop of CIDEr-D and SPICE for ReCap$_{\text{Prompts+Tokens}}$ due to the lack of information. Surprisingly, the worst method in terms of CIDEr-D and SPICE (ReCap$_{\text{Tokens}}^-$) achieves higher scores in terms of CLIP-S and CLIP-RS than our best method (ReCap$_{\text{Captions}}$). We observed similar behaviour when using CLIP-S as a metric for filtering synthetic captions in our self-improvement loop. Narratives contain very tailored descriptions for images and represent a distribution mismatch with original reference captions used for evaluation on the test set. Hence, we observe decreased performance for narratives.

## 3.5 ANALYSIS OF CLIP-SCORE AND REFCLIP-SCORE

We found that CLIP indeed often assigns unusually high scores to low-quality captions produced by ReCap$_{\text{Tokens}}^-$. We further investigate this phenomenon by a qualitative evaluation of captions generated

---

[4]We follow the prompting strategy of `https://github.com/openai/CLIP/blob/main/notebooks/Prompt_Engineering_for_ImageNet.ipynb`

[5]We also perform common preprocessing steps, such as stop-word removal and deduplication.

Table 4: Comparison of different language supervisions ranging from token-level to narratives on the MS-COCO test split. We report mean and standard error for all metrics except for CLIP-S and CLIP-RS where the standard error is negligible. − indicates method does not use the linear mapping.

| Method | CIDEr-D | SPICE | CLIP-S | CLIP-RS |
|---|---|---|---|---|
| ReCap$^-_{\text{Tokens}}$ | $15.4 \pm 0.3$ | $5.5 \pm 0.1$ | 75.5 | 78.5 |
| ReCap$^-_{\text{Prompts+Tokens}}$ | $17.5 \pm 0.3$ | $6.2 \pm 0.1$ | 75.8 | 78.7 |
| ReCap$^-_{\text{Captions}}$ | $79.6 \pm 0.9$ | $18.1 \pm 0.1$ | 78.1 | 80.0 |
| ReCap$^-_{\text{AllCaps}}$ | $80.0 \pm 0.9$ | $17.6 \pm 0.1$ | 73.5 | 77.5 |
| ReCap$^-_{\text{LN}}$ | $41.2 \pm 0.7$ | $11.6 \pm 0.1$ | 69.4 | 75.0 |
| ReCap$_{\text{Tokens}}$ | $46.9 \pm 0.6$ | $13.8 \pm 0.1$ | 73.3 | 77.4 |
| ReCap$_{\text{Prompts+Tokens}}$ | $41.1 \pm 0.5$ | $12.4 \pm 0.1$ | 72.3 | 76.8 |
| ReCap$_{\text{Captions}}$ | $103.3 \pm 1.0$ | $20.8 \pm 0.1$ | 74.6 | 78.2 |
| ReCap$_{\text{AllCaps}}$ | $89.5 \pm 0.9$ | $19.0 \pm 0.1$ | 73 | 77.2 |
| ReCap$_{\text{LN}}$ | $42.7 \pm 0.6$ | $12.1 \pm 0.1$ | 67.9 | 74.1 |

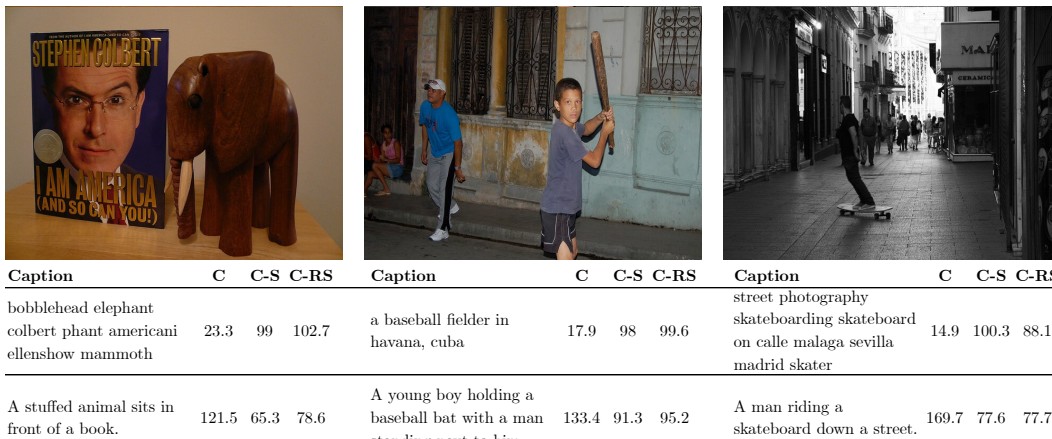

| Caption | C | C-S | C-RS | Caption | C | C-S | C-RS | Caption | C | C-S | C-RS |
|---|---|---|---|---|---|---|---|---|---|---|---|
| bobblehead elephant colbert phant americani ellenshow mammoth | 23.3 | 99 | 102.7 | a baseball fielder in havana, cuba | 17.9 | 98 | 99.6 | street photography skateboarding skateboard on calle malaga sevilla madrid skater | 14.9 | 100.3 | 88.1 |
| A stuffed animal sits in front of a book. | 121.5 | 65.3 | 78.6 | A young boy holding a baseball bat with a man standing next to him. | 133.4 | 91.3 | 95.2 | A man riding a skateboard down a street. | 169.7 | 77.6 | 77.7 |

Figure 3: Sample images, CIDEr-D (C), CLIP-score (C-S), and RefCLIP-score (C-RS) for captions generated via ReCap (bottom) and ReCap$_{\text{Tokens}}$.

by either of the two. In the extreme case, CLIP even assigns higher scores to a bag of words than to an actual caption. We show some examples for low-quality captions and measured CIDEr-D, CLIP-S, and CLIP-RS in Figure 3.

The image on the left shows a caption consisting of a bag of words. CLIP-S is higher for the bag-of-words caption (top) than for the valid caption on the bottom. Further, CLIP-RS does not correct for this artifact, but is even higher than CLIP-S. This is due to the fact that CLIP-RS only penalizes a generated caption if the maximum cosine similarity of the references is smaller than CLIP-S, or if CLIP-S is generally low. However, as long as some semantically related concept appears in the generated caption, CLIP-S tends to be high. Thus, both scores only give a measure as to whether or not a caption is semantically related to an image. For the image on the right, CLIP-RS corrects the low quality score a bit, but this is only due to the fact, that lower similarity is assigned to the reference captions for this image. Finally, in the middle image CLIP-S and CLIP-RS are affected by hallucinated content such as "in havana, cuba" and assigns a higher score than to the caption on the bottom which attains a high CIDEr-D score. We provide more of these examples in Appendix C.

To further investigate this phenomenon, we analyze the correlations between all metrics. Usually, one would expect quite strong positive correlations among commonly used metrics, i.e., the higher quality of the caption, the higher the score for the different metrics. Figure 4 shows the pearson correlation for all metrics when evaluating our best ReCap setting on the MS-COCO test split. N-gram based metrics, such as CIDEr-D, ROUGE-L, and BLEU@4 strongly correlate with each other, while there

is only a slight positive correlation with CLIP-based metrics. Further, perhaps surprisingly, SPICE is almost entirely decorrelated from all other metrics. This is because it is not based on n-grams but uses semantic scene graphs for evaluation (Anderson et al., 2016). Still, SPICE correctly assigns lower scores to methods that produce low-quality captions, as shown in Table 4. Finally, CLIP-S and CLIP-RS strongly correlate with each other, while being weakly correlated to all the other metrics.

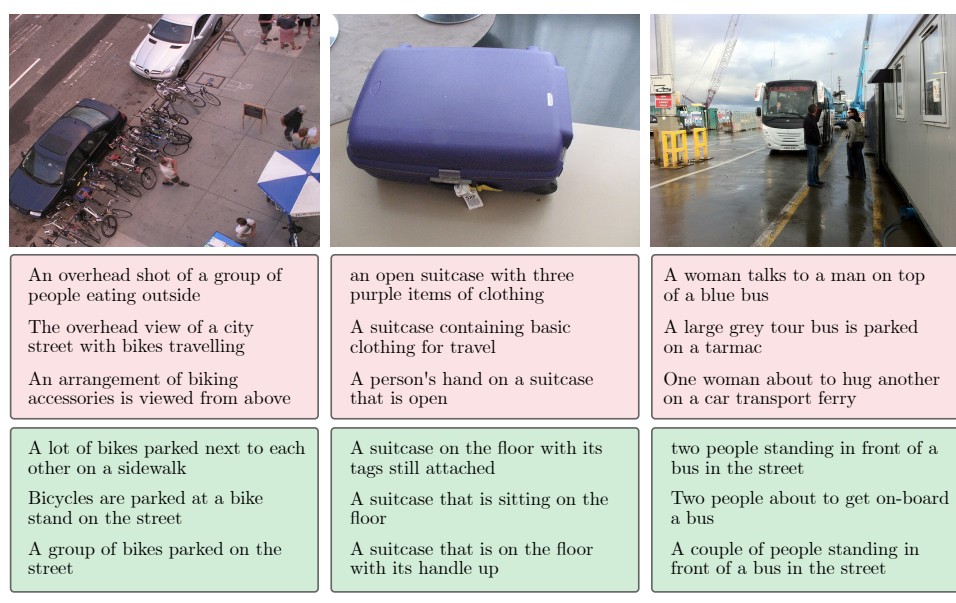

Figure 5: Sample images with retrieved captions with and without our mapping for closing the modality gap. We show three of the closest captions to an image. Images are taken from the MS-COCO validation set.

# 4 RELATED WORK

**Image Captioning** The task of image captioning has been widely considered in the literature (Stefanini et al., 2023; Tan & Bansal, 2019; Zhou et al., 2019; Yao et al., 2018; Xu et al., 2015; Li et al., 2020; Fang et al., 2015; Chen & Zitnick, 2014; Anderson et al., 2018). Early works employed pretrained image classification models Chen & Zitnick (2014); Chen et al. (2017); Fang et al. (2015); Xu et al. (2015) or domain specific object detectors (Ren et al., 2017). Further, attention mechanisms were deployed to allow attending to different visual cues (Anderson et al., 2018; Xu et al., 2015; Chen et al., 2017). For mapping visual features to text several works used the LSTM architecture (Chen et al., 2018; Vinyals et al., 2015; Wang et al., 2017), or the Transformer architecture (Herdade et al., 2019; Yang et al., 2019; Dosovitskiy et al., 2021; Liu et al., 2021). Then the focus shifted towards pretraining on vast datasets of paired image-text data and subsequent finetuning for image captioning (Li et al., 2020; Tan & Bansal, 2019; Zhang et al., 2021; Zhou et al., 2019; Wang et al., 2021; 2022).

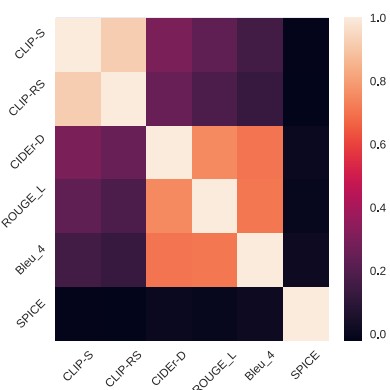

Figure 4: Pearson correlation between commonly used image captioning metrics for captions generated via ReCap on the MS-COCO test set.

**Transferring visual input to a pretrained LM** Due to the rapid evolution of LMs, a plethora of works proposed to bootstrap their generation capabilities and condition them on visual input. One way

to transfer visual inputs to a LM is via various forms of cross-attention between pretrained unimodal models (Luo et al., 2023; Lu et al., 2019; Ramos et al., 2022; Alayrac et al., 2022; Yang et al., 2023b; Koh et al., 2023; Li et al., 2022). Another way to fuse visual input to a LM is to only train a mapping network between images and the LM input space (Mokady et al., 2021; Zhu et al., 2023; Merullo et al., 2022; Li et al., 2023a; Tsimpoukelli et al., 2021; Scialom et al., 2020; Driess et al., 2023; Liu et al., 2023; Huang et al., 2023) Finally, other approaches rely on a semantic alignment of image and text modalities via contrastive learning (Radford et al., 2021; Li et al., 2021).

**Lightweight Image Captioning**   Eichenberg et al. (2022); Zhang et al. (2023); Gao et al. (2023) interleave a pretrained LM with adapter layers (Rebuffi et al., 2018) conditioned on images. Other works fuse visual input into a LM by training parameter-efficient cross-attention modules (Luo et al., 2023), or a mapping network between embedding spaces (Mokady et al., 2021; Merullo et al., 2022). More recently, Ramos et al. (2022) proposed retrieval augmentation leveraging a pretrained VLM combined with a cross-attention mechanism trained end-to-end. (Ramos et al., 2023) uses retrieval augmentation to obtain multilingual prompts which enables generation in a certain target language. All of these approaches optimize the mapping from image space to the embedding space or hidden space of the pretrained LM. Our work aims at grounding images to captions from the training set in the joint embedding space of CLIP to bridge the modality gap. This way, we enhance the retrieval component and only need to provide retrieved captions in the form of text to the LM.

**Bridging the modality gap**   Similar to our approach, Ouali et al. (2023) use orthogonal procrustes to mitigate the modality gap of CLIP-like models for few-shot classification. Our method uses an ordinary least squares mapping, to enhance retrieval augmented text generation from images. To avoid the need for bridging the modality gap, other works consider image captioning using only text data by training a text decoder for CLIP-style models (Li et al., 2023b; Nukrai et al., 2022; Yu et al., 2022; Wang et al., 2023; Gu et al., 2022). However, at test time these approaches still receive images as inputs, thus are still affected by the modality gap. Our approach mitigates this issue by grounding images to captions in a given dataset via a linear mapping. Other approaches adapt the pretraining objective in order to achieve a better alignment of image and text modalities in the joint embedding space (Fürst et al., 2022; Goel et al., 2022; Humer et al., 2023). While these methods effectively close the modality gap, they were trained on smaller datasets than CLIP. Therefore, we still use CLIP as our retrieval system and apply the linear mapping for task-specific grounding.

## 5   DISCUSSION AND LIMITATIONS

**Datastore dependency**   Usually image captioning pipelines are trained end-to-end on a training collection of image-text pairs. Contrary, ReCap only trains the retrieval mechanism by grounding training images to corresponding captions. This results in an efficient alternative to end-to-end training. The drawback of ReCap is that embeddings for training captions need to be stored explicitly, which increases the memory footprint. In practice, this did not result in substantial overhead though, since the number of training captions was moderate. In case of large-scale datasets, faiss provides the possibility to compress the datastore and, in turn, reduce the memory footprint. Further, the datastore could potentially be augmented with additional captions from different training sets as shown by (Ramos et al., 2022).

**What metrics should be reported?**   Usually the usefulness of a metric is evaluated by measuring correlation with human judgement. Humans generally tend to prefer *correctness* over *specificity* in image captions (Rohrbach et al., 2018; 2017). While CLIP-score exhibits strong correlation with human judgements and is robust to object hallucination (Hessel et al., 2021), we found that it rather evaluates for semantic relatedness than correctness. Our findings are corroborated by other works that have found a severe lack of order sensitivity and compositionality in CLIP representations (Yüksekgönül et al., 2023; Zhao et al., 2022; Thrush et al., 2022). Therefore, we recommend to (i) report multiple metrics, (ii) always incorporate rule-based metrics such as CIDEr-D or SPICE, and (iii) opt for metrics that rely on semantic similarity between candidate and reference captions along with visual information (Jiang et al., 2019; Lee et al., 2020). In case of large gaps between metrics, as in (Zeng et al., 2023), we recommend to conduct a thorough qualitative analysis to ensure caption quality.

**Synthetic data**   Our proposed self-improvement loop relies on synthetic data generated by the generative LM. Recent works have shown the benefits of adding synthetic data to existing datasets (Gülçehre et al., 2023; Yang et al., 2023a; Lin et al., 2023). However, other recent work has shown that training on synthetic data can result in the so-called *model-collapse*, where the tails of the training distribution shrink over time (Shumailov et al., 2023). Since we iteratively add synthetic captions to our dataset, this concerns our self-improvement loop as well. However, we only add captions to our dataset that yield high scores to certain metrics that capture similarity to human references. Future work should investigate model-collapse in our setup and whether it is responsible for the slight decrease in certain metrics.

**Training time**   While training is very efficient, the self-improvement step requires much more compute, because we need to iterate over the entire training corpus to generate synthetic captions. For datasets such as MS-COCO this process took approximately 15 hours for one iteration. While certain metrics can be improved with this procedure it is essentially a performance-vs-complexity trade-off. If best performance is not the main goal, we recommend to either perform one iteration of self-improvement, or to neglect it entirely. We believe, however, that our self-improvement loop can be useful for low-resource data settings, which we aim to investigate in the future.

## 6   CONCLUSION

We introduced an efficient method to bridge the prevalent modality gap in pretrained VLMs for the task of image captioning. To this end, we compute a linear mapping between corresponding image-caption pairs provided by existing datasets, such as MS-COCO or Flickr30k. The linear mapping can be computed in closed form on CPU. Given an image, we apply our mapping and retrieve the closest captions of the trainng set. Along with an instruction, these captions serve as input to a generative LM to generate new captions. Moreover, we propose a novel self-improvement loop to iteratively refine the mapping based on captions bootstrapped by the LM. We only keep synthetic captions that attain a high score for a metric of interest and add these to the training set for the lightweight mapping. This way we can further improve on certain captioning metrics. Moreover, we find that reference-free metrics, such as CLIP-score can be fooled by hallucinated contents or even a simple bag of words. Our method attains competitive performance to existing lightweight image captioning methods. Finally, our mapping enables the use of relatively small LMs for image captioning. Thus, we make image captioning more accessible for users with limited resources.

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

## A HYPERPARAMETER SEARCH

**Effect of different vision encoders** We investigate the effect of different vision encoders on the captioning performance. In this regard, we compare all publicly available encoder variants of CLIP, which comprise ViT-based (Dosovitskiy et al., 2021), as well as resnet-based (He et al., 2016) architectures. We observe a significant improvement in captioning performance when using a resnet encoder as shown in Table 5.

**Different decoding strategies** As illustrated by (Holtzman et al., 2020), the decoding strategy substantially affects human approval of generated captions. Therefore, we evaluate different decoding strategies, including greedy decoding, sampling, top-k sampling, and nucleus sampling. For the sampling-based strategies we follow hyperparameter settings from (Holtzman et al., 2020). The results for the different decoding schemes are shown in Table 6. Surprisingly, we found that ReCap generates the best captions using greedy decoding. Other sampling strategies tend to diverge from captions provided in the context, which results in lower CIDEr-D scores. Generally, sampling-based decoding results in more variety in generated captions. We can achieve a similar effect by permuting our prompt ordering and using greedy decoding while avoiding divergence of generated captions.

Table 5: Search over all publicly available CLIP vision encoder backbones evaluated on the MS-COCO validation set. We report mean and standard error for all settings. $|\theta|$ denotes the number of trainable parameters.

| Vision Encoder | BLEU@1 | BLEU@4 | ROUGE-L | CIDEr-D | SPICE | $|\theta|$ |
|---|---|---|---|---|---|---|
| RN50 | $75.3 \pm 0.2$ | $27.9 \pm 0.3$ | $56.1 \pm 0.2$ | $95.9 \pm 0.9$ | $19.3 \pm 0.1$ | 1 M |
| RN101 | $74.9 \pm 0.2$ | $27.9 \pm 0.3$ | $56.0 \pm 0.2$ | $95.7 \pm 0.9$ | $19.2 \pm 0.1$ | 262 K |
| RN50x4 | $75.6 \pm 0.2$ | $29.0 \pm 0.3$ | $56.7 \pm 0.2$ | $99.5 \pm 0.9$ | $19.8 \pm 0.1$ | 410 K |
| RN50x16 | $76.4 \pm 0.2$ | $29.5 \pm 0.3$ | $57.0 \pm 0.2$ | $101.9 \pm 0.9$ | $20.1 \pm 0.1$ | 590 K |
| RN50x64 | $77.5 \pm 0.2$ | $30.7 \pm 0.4$ | $57.9 \pm 0.2$ | $105.8 \pm 1.0$ | $20.8 \pm 0.1$ | 1 M |
| ViT-B/32 | $75.2 \pm 0.2$ | $28.0 \pm 0.3$ | $56.1 \pm 0.2$ | $96.1 \pm 0.9$ | $19.2 \pm 0.1$ | 262 K |
| ViT-B/16 | $76.2 \pm 0.2$ | $29.1 \pm 0.4$ | $56.7 \pm 0.2$ | $100.4 \pm 1.0$ | $19.7 \pm 0.1$ | 262 K |
| ViT-L/14 | $77.0 \pm 0.2$ | $30.2 \pm 0.4$ | $57.4 \pm 0.2$ | $104.2 \pm 1.0$ | $20.3 \pm 0.1$ | 590 K |
| ViT-L/14@336px | $77.2 \pm 0.2$ | $30.1 \pm 0.4$ | $57.4 \pm 0.2$ | $104.3 \pm 0.9$ | $20.4 \pm 0.1$ | 590 K |

Table 6: Search over different decoding paradigms for captioning on the MS-COCO validation set. We report mean and standard error for all settings

| Decoding | BLEU@1 | BLEU@4 | ROUGE-L | CIDEr-D | SPICE |
|---|---|---|---|---|---|
| Sampling | $52.6 \pm 0.3$ | $12.3 \pm 0.2$ | $42.4 \pm 0.2$ | $47.5 \pm 0.7$ | $14.2 \pm 0.1$ |
| Topk | $54.0 \pm 0.3$ | $12.7 \pm 0.2$ | $43.2 \pm 0.2$ | $50.0 \pm 0.7$ | $14.7 \pm 0.1$ |
| Nucleus | $64.7 \pm 0.2$ | $18.6 \pm 0.3$ | $49.1 \pm 0.2$ | $71.7 \pm 0.8$ | $17.5 \pm 0.1$ |
| Greedy | $77.5 \pm 0.2$ | $30.7 \pm 0.4$ | $57.9 \pm 0.2$ | $105.8 \pm 1.0$ | $20.8 \pm 0.1$ |

**Language Model Scale** We evaluate FLAN-T5 model sizes of 250M, 720M, 3B, and 11B scales. Further, we include decoder-only LMs, such as GPT-J (Wang & Komatsuzaki, 2021) and Llama 7B (Touvron et al., 2023). The results can be observed in Table 7. Our results show that there is not much performance gain going from FLAN-T5-LARGE to FLAN-T5-XXL. We believe that the summarization task itself is being solved in a very good way by the large version of FLAN-T5 already. Thus, the performance-complexity trade-off imposed by using larger models is not worth considering. Surprisingly, even the small variant of FLAN-T5 reaches a CIDEr-D score above 90, outperforming the base version, however we do not have a good intuition why that is the case. For the decoder-only LMs we use the same prompting strategy as (Ramos et al., 2022). Our results show that decoder-only LMs generally perform worse than encoder-decoder ones. A possible reason for that is the lack of bidirectionality in the encoder, which is inherent to encoder-decoder models. We found that decoder-only models are generally more sensitive to prompt ordering. Also, perhaps surprisingly, GPT-J (Wang & Komatsuzaki, 2021) outperforms the recently proposed Llama (Touvron et al., 2023), which reaches performance on-par with GPT-2 (Radford et al., 2018). Overall there is a clear trend that larger models do not necessarily lead to better performance. The most performance gains can be achieved by focusing on LMs that are better suited for the task at hand.

Usually we would provide the exemplars in the prompt from most-similar to least similar, i.e. the least similar prompt is the most recent in the context. However, one may think the exact opposite ordering might lead to better captioning performance, since the LM might exhibit a form of recency bias. Hence, we provide results for the worst-to-best ordering in Table 8. Indeed, we found that different ordering of exemplars in the prompt leads to different results. Ordering from worst-to-best, i.e. most similar exemplars appear more recently, leads to a significant improvement in CIDEr-D score. This corroborates findings of (Zhao et al., 2021) that LMs are prone to prompt ordering. In our case this is illustrated in a recency bias of the used FLAN-T5.

Further, we search over different values for our hyperparameters $k$ and $l$ on the MS-COCO and on the Flickr30k validation sets. We report results are in Table 9 and Table 10 for MS-COCO, and Flickr30k, respectively.

Table 7: Comparison of different language models on the MS-COCO validation set. We report mean and standard error for all settings.

| Model | BLEU@1 | BLEU@4 | ROUGE-L | CIDEr-D | SPICE |
|---|---|---|---|---|---|
| | | | Encoder-Decoder | | |
| FLAN-T5-SMALL | $57.3 \pm 0.3$ | $20.9 \pm 0.3$ | $53.9 \pm 0.2$ | $89.8 \pm 0.9$ | $20.3 \pm 0.1$ |
| FLAN-T5-BASE | $60.2 \pm 0.2$ | $22.2 \pm 0.3$ | $54.7 \pm 0.2$ | $92.5 \pm 0.9$ | $20.5 \pm 0.1$ |
| FLAN-T5-LARGE | $77.5 \pm 0.2$ | $30.7 \pm 0.4$ | $57.9 \pm 0.2$ | $105.8 \pm 1.0$ | $20.8 \pm 0.1$ |
| FLAN-T5-XL | $76.1 \pm 0.2$ | $29.5 \pm 0.4$ | $56.8 \pm 0.2$ | $103.1 \pm 0.9$ | $20.4 \pm 0.1$ |
| FLAN-T5-XXL | $64.0 \pm 0.3$ | $23.3 \pm 0.3$ | $54.6 \pm 0.2$ | $94.4 \pm 0.1$ | $- \pm 0.1$ |
| | | | Decoder-only | | |
| GPT-2 | $67.8 \pm 0.3$ | $24.6 \pm 0.3$ | $49.6 \pm 0.2$ | $87.7 \pm 0.9$ | $19.3 \pm 0.1$ |
| GPT-J 6B | $71.1 \pm 0.3$ | $27.1 \pm 0.3$ | $51.2 \pm 0.2$ | $93.6 \pm 0.9$ | $19.5 \pm 0.1$ |
| Llama 7B | $63.5 \pm 0.3$ | $23.9 \pm 0.3$ | $49.6 \pm 0.2$ | $87.9 \pm 0.9$ | $19.2 \pm 0.1$ |

Table 8: Comparison of different orderings for exemplars in the prompt on the MS-COCO validation set. We report mean and standard error for all settings.

| Ordering | BLEU@1 | BLEU@4 | ROUGE-L | CIDEr-D | SPICE |
|---|---|---|---|---|---|
| worst-to-best | $77.5 \pm 0.2$ | $30.7 \pm 0.4$ | $57.9 \pm 0.2$ | $105.8 \pm 1.0$ | $20.8 \pm 0.1$ |
| best-to-worst | $77.2 \pm 0.2$ | $30.6 \pm 0.4$ | $57.7 \pm 0.2$ | $104.6 \pm 0.9$ | $20.7 \pm 0.1$ |

## B    POTENTIAL SOCIETAL IMPACT

Our method uses foundation models, which were trained on uncurated datasets crawled from the web. Therefore, these models readily reflect prejudices and biases found on the web. Consequently, our proposed captioning system might also bear these shortcomings. In the worst case, this could lead to our method producing inappropriate or even harmful contents. Moreover, generative LMs as used by our method are known to be very sensitive to prompting Zhao et al. (2021) and can therefore be misused if a user gets to determine certain prompts. However, our method is also very low in complexity and makes caption generation more accessible to researchers suffering from hardware constraints. Due to the low number of parameters and the simple training procedure it can efficiently be adapted to different domains.

## C    ADDITIONAL QUALITATIVE ANALYSIS

We provide additional examples for the susceptibility of CLIP-score to hallucinated contents in Figure 6. The captions from ReCap$_{\text{Tokens}}$ contain plenty of hallucinated content, e.g. the imaginary person "clayton cha", asses grazing along with zebras, a "mii peripheral", the "icelandic shetland", or "halt homestead". We observe that CLIP assigns very high scores to such content, even if the generated caption is not even syntactically valid, e.g. bottom left image. Although CLIP-RS includes reference captions, it only corrects the score for the generated caption if the maximum cosine similarity between references and image is lower than the CLIP-score. Contrary, if the maximum cosine similarity between image and references is higher than the CLIP-score, CLIP-RS will also be higher. On the bottom right, for example, the CLIP-S for the valid caption is reduced because the maximum similarity to reference captions is lower, although there is a high overlap in terms of n-gram overlap, indicated by the CIDEr-D score.

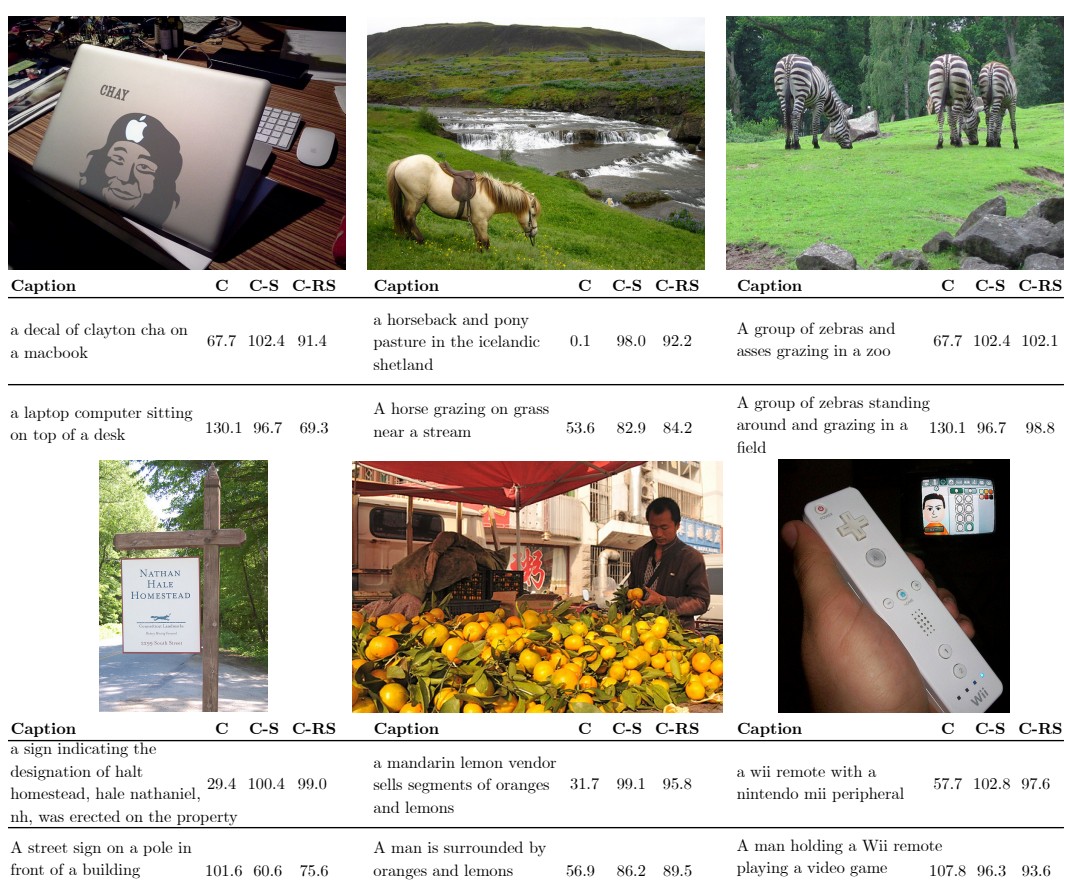

| Caption | C | C-S | C-RS |
|---|---|---|---|
| a decal of clayton cha on a macbook | 67.7 | 102.4 | 91.4 |
| a laptop computer sitting on top of a desk | 130.1 | 96.7 | 69.3 |

| Caption | C | C-S | C-RS |
|---|---|---|---|
| a horseback and pony pasture in the icelandic shetland | 0.1 | 98.0 | 92.2 |
| A horse grazing on grass near a stream | 53.6 | 82.9 | 84.2 |

| Caption | C | C-S | C-RS |
|---|---|---|---|
| A group of zebras and asses grazing in a zoo | 67.7 | 102.4 | 102.1 |
| A group of zebras standing around and grazing in a field | 130.1 | 96.7 | 98.8 |

| Caption | C | C-S | C-RS |
|---|---|---|---|
| a sign indicating the designation of halt homestead, hale nathaniel, nh, was erected on the property | 29.4 | 100.4 | 99.0 |
| A street sign on a pole in front of a building | 101.6 | 60.6 | 75.6 |

| Caption | C | C-S | C-RS |
|---|---|---|---|
| a mandarin lemon vendor sells segments of oranges and lemons | 31.7 | 99.1 | 95.8 |
| A man is surrounded by oranges and lemons | 56.9 | 86.2 | 89.5 |

| Caption | C | C-S | C-RS |
|---|---|---|---|
| a wii remote with a nintendo mii peripheral | 57.7 | 102.8 | 97.6 |
| A man holding a Wii remote playing a video game | 107.8 | 96.3 | 93.6 |

Figure 6: Images from the MS-COCO validation set, along with generated captions from ReCap (top line) and ReCap$_{\text{Tokens}}$ (bottom line), along with CIDEr-D (C), CLIP-score (C-S), and RefCLIP-score (C-RS).

Table 9: Hyperparameter Search for $k$ on the MS-COCO validation set for different levels of language abstraction using our semantic mapping computed via OLS. We report mean and standard error for all settings. We select the best k according to CIDEr-D score.

| $k$ | BLEU@1 | BLEU@4 | ROUGE-L | CIDEr-D | SPICE |
|---|---|---|---|---|---|
| | | | Single Captions | | |
| 10 | $77.5 \pm 0.2$ | $30.3 \pm 0.4$ | $57.6 \pm 0.2$ | $105.3 \pm 1.0$ | $20.9 \pm 0.1$ |
| 11 | $77.6 \pm 0.2$ | $30.5 \pm 0.4$ | $57.7 \pm 0.2$ | $105.4 \pm 1.0$ | $21.0 \pm 0.1$ |
| 12 | $77.6 \pm 0.2$ | $30.5 \pm 0.4$ | $57.7 \pm 0.2$ | $105.4 \pm 1.0$ | $21.0 \pm 0.1$ |
| 13 | $77.5 \pm 0.2$ | $30.5 \pm 0.4$ | $57.6 \pm 0.2$ | $105.3 \pm 1.0$ | $20.8 \pm 0.1$ |
| 14 | $77.4 \pm 0.2$ | $30.5 \pm 0.4$ | $57.8 \pm 0.2$ | $105.4 \pm 1.0$ | $20.8 \pm 0.1$ |
| 15 | $77.5 \pm 0.2$ | $30.7 \pm 0.4$ | $57.9 \pm 0.2$ | $105.8 \pm 1.0$ | $20.8 \pm 0.1$ |
| 16 | $77.4 \pm 0.2$ | $30.4 \pm 0.4$ | $57.8 \pm 0.2$ | $105.3 \pm 1.0$ | $20.9 \pm 0.1$ |
| 17 | $77.3 \pm 0.2$ | $30.5 \pm 0.4$ | $57.7 \pm 0.2$ | $105.3 \pm 1.0$ | $20.9 \pm 0.1$ |
| 18 | $77.4 \pm 0.2$ | $30.6 \pm 0.4$ | $57.7 \pm 0.2$ | $105.5 \pm 1.0$ | $20.9 \pm 0.1$ |
| 19 | $77.4 \pm 0.2$ | $30.5 \pm 0.4$ | $57.7 \pm 0.2$ | $105.6 \pm 1.0$ | $20.9 \pm 0.1$ |
| 20 | $77.5 \pm 0.2$ | $30.6 \pm 0.4$ | $57.8 \pm 0.2$ | $105.5 \pm 1.0$ | $21.0 \pm 0.1$ |
| | | | All Captions | | |
| 1 | $72.7 \pm 0.2$ | $24.8 \pm 0.3$ | $53.9 \pm 0.2$ | $87.0 \pm 0.9$ | $18.0 \pm 0.1$ |
| 2 | $73.7 \pm 0.2$ | $26.4 \pm 0.3$ | $54.7 \pm 0.2$ | $90.8 \pm 0.9$ | $18.2 \pm 0.1$ |
| 3 | $74.0 \pm 0.2$ | $26.4 \pm 0.3$ | $54.8 \pm 0.2$ | $91.0 \pm 0.9$ | $18.2 \pm 0.1$ |
| 4 | $74.0 \pm 0.2$ | $26.6 \pm 0.3$ | $55.0 \pm 0.2$ | $91.3 \pm 0.9$ | $18.5 \pm 0.1$ |
| 5 | $74.0 \pm 0.2$ | $26.9 \pm 0.3$ | $55.1 \pm 0.2$ | $91.6 \pm 0.9$ | $18.4 \pm 0.1$ |
| | | | Localized Narratives | | |
| 1 | $55.3 \pm 0.3$ | $11.7 \pm 0.2$ | $43.1 \pm 0.2$ | $45.4 \pm 0.6$ | $11.9 \pm 0.1$ |
| 2 | $54.3 \pm 0.3$ | $11.8 \pm 0.2$ | $43.0 \pm 0.2$ | $48.0 \pm 0.7$ | $13.2 \pm 0.1$ |
| 3 | $53.8 \pm 0.3$ | $12.3 \pm 0.2$ | $43.0 \pm 0.2$ | $50.9 \pm 0.7$ | $14.0 \pm 0.1$ |
| 4 | $53.0 \pm 0.3$ | $12.1 \pm 0.2$ | $42.7 \pm 0.2$ | $51.7 \pm 0.7$ | $14.3 \pm 0.1$ |
| 5 | $52.5 \pm 0.3$ | $12.0 \pm 0.2$ | $42.6 \pm 0.2$ | $52.6 \pm 0.7$ | $14.4 \pm 0.1$ |
| 6 | $52.0 \pm 0.3$ | $12.3 \pm 0.2$ | $42.6 \pm 0.2$ | $53.1 \pm 0.7$ | $14.6 \pm 0.1$ |

---

**Algorithm 2** Self-improvement loop

**Require:** CLIP vision encoder $\phi(\cdot)$, CLIP text encoder $\psi(\cdot)$, Training set $\mathcal{D}_{\text{Train}} = \{(\boldsymbol{x}_i, \boldsymbol{c}_i)\}$, Validation set $\mathcal{D}_{\text{Val}} = \{(\boldsymbol{x}_j)\}$, Hyperparameter $k$, Language Model $\text{LM}(\cdot)$, Prompt $\mathcal{P}$, Number of iterations $n$

$\{(\boldsymbol{f}_i, \boldsymbol{e}_i)\}_{i=1}^{|\mathcal{D}_{\text{Train}}|} \leftarrow \phi(\boldsymbol{x}_i), \psi(\boldsymbol{s}_i) \text{ for } (\boldsymbol{x}_i, \boldsymbol{c}_i) \in \mathcal{D}_{\text{Train}}$      ▷ Embed training set
$\boldsymbol{W} \leftarrow \texttt{fit\_linear}(\{(\boldsymbol{f}_i, \boldsymbol{e}_i)\})$      ▷ Pre-compute linear mapping
$\mathcal{B} \leftarrow \{\boldsymbol{e}_i\}$      ▷ Initialize datastore with training captions

$\bar{m} \leftarrow \texttt{evaluate}(\mathcal{D}_{\text{Val}}, \phi, \boldsymbol{W}, \text{LM}, \mathcal{B})$      ▷ Evaluate on the validation set
$\textbf{for}\ \_ \ \textbf{in}\ \texttt{range}(n)$      ▷ Run self-improvement for $n$ iterations
    $\{\mathcal{E}_i\} \leftarrow \texttt{topk}(\{\boldsymbol{W}\boldsymbol{f}_i\}, \mathcal{B}, k)$      ▷ Retrieve captions from datastore for each training image
    $\{\mathcal{S}_i\} \leftarrow \text{LM}(\texttt{concat}(\mathcal{P} + \mathcal{E}_i)$      ▷ Generate new captions for each trainig image
    $\{\mathcal{S}_i\} \leftarrow \texttt{filter\_and\_deduplicate}(\{\mathcal{S}_i\}, \bar{m})$      ▷ Filter captions according to $\bar{m}$
    $\{\boldsymbol{e}_l\}_{l=1}^{|\mathcal{D}_{\text{Train}}|} \leftarrow \psi(\boldsymbol{s}_l) \text{ for } (\boldsymbol{s}_l) \in \mathcal{S}$      ▷ Embed new synthetic captions
    $\mathcal{B} \leftarrow \mathcal{B} \cup \{(\boldsymbol{e}_l)\}$      ▷ Add synthetic captions to datastore
    $\{(\boldsymbol{f}_i, \boldsymbol{e}_i, \boldsymbol{e}_l)\}_{i=1, l=1}^{|\mathcal{D}_{\text{Train}}|} \leftarrow \boldsymbol{e}_l \text{ for } (\boldsymbol{e}_l) \in \{(\boldsymbol{e}_l)\}$      ▷ Augment training set
    $\boldsymbol{W} \leftarrow \texttt{fit\_linear}(\{(\boldsymbol{f}_i, \boldsymbol{e}_i, \boldsymbol{e}_l)\})$      ▷ Re-compute $\boldsymbol{W}$
    $\bar{m} \leftarrow \texttt{evaluate}(\mathcal{D}_{\text{Val}}, \phi, \boldsymbol{W}, \text{LM}, \mathcal{B})$      ▷ Update $\bar{m}$

Table 10: Hyperparameter Search for $k$ on the Flickr30k validation set for different levels of language abstraction using our semantic mapping computed via OLS. For tokens as targets we additionally search over the number of random permutations $l$. We report mean and standard error for all settings.

| $k$ | BLEU@1 | BLEU@4 | ROUGE-L | CIDEr-D | SPICE |
|---|---|---|---|---|---|
| | | | Single Captions | | |
| 10 | $74.9 \pm 0.5$ | $26.5 \pm 0.7$ | $54.6 \pm 0.4$ | $63.9 \pm 1.9$ | $15.5 \pm 0.3$ |
| 11 | $74.7 \pm 0.5$ | $26.0 \pm 0.7$ | $54.3 \pm 0.4$ | $64.0 \pm 1.9$ | $15.5 \pm 0.3$ |
| 12 | $74.4 \pm 0.5$ | $26.2 \pm 0.7$ | $54.5 \pm 0.4$ | $64.3 \pm 1.9$ | $15.5 \pm 0.3$ |
| 13 | $74.2 \pm 0.5$ | $26.3 \pm 0.7$ | $54.6 \pm 0.4$ | $64.6 \pm 1.9$ | $15.2 \pm 0.3$ |
| 14 | $74.5 \pm 0.5$ | $26.2 \pm 0.7$ | $54.3 \pm 0.4$ | $64.4 \pm 1.9$ | $15.5 \pm 0.3$ |
| 15 | $74.2 \pm 0.5$ | $26.2 \pm 0.7$ | $54.4 \pm 0.4$ | $64.6 \pm 1.9$ | $15.6 \pm 0.3$ |
| 16 | $74.8 \pm 0.5$ | $26.8 \pm 0.7$ | $54.6 \pm 0.4$ | $65.0 \pm 1.9$ | $15.8 \pm 0.3$ |
| 17 | $74.5 \pm 0.5$ | $26.6 \pm 0.7$ | $54.7 \pm 0.4$ | $64.7 \pm 1.9$ | $15.7 \pm 0.3$ |
| | | | All Captions | | |
| 1 | $65.8 \pm 0.5$ | $20.3 \pm 0.7$ | $49.8 \pm 0.4$ | $48.7 \pm 1.8$ | $13.4 \pm 0.3$ |
| 2 | $67.9 \pm 0.5$ | $21.5 \pm 0.7$ | $50.5 \pm 0.5$ | $52.2 \pm 1.8$ | $13.9 \pm 0.3$ |
| 3 | $68.1 \pm 0.5$ | $22.0 \pm 0.7$ | $51.0 \pm 0.4$ | $53.2 \pm 1.9$ | $13.7 \pm 0.3$ |
| 4 | $69.6 \pm 0.5$ | $23.0 \pm 0.7$ | $51.4 \pm 0.4$ | $54.4 \pm 1.9$ | $14.1 \pm 0.3$ |
| 5 | $69.0 \pm 0.5$ | $23.0 \pm 0.7$ | $51.3 \pm 0.4$ | $54.5 \pm 1.9$ | $14.2 \pm 0.3$ |
| | | | Localized Narratives | | |
| 1 | $54.2 \pm 0.6$ | $9.0 \pm 0.4$ | $40.4 \pm 0.4$ | $24.4 \pm 1.3$ | $8.1 \pm 0.2$ |
| 2 | $52.6 \pm 0.6$ | $8.6 \pm 0.4$ | $39.3 \pm 0.4$ | $23.3 \pm 1.1$ | $8.4 \pm 0.2$ |
| 3 | $52.5 \pm 0.6$ | $9.5 \pm 0.4$ | $39.6 \pm 0.4$ | $25.4 \pm 1.2$ | $8.9 \pm 0.2$ |
| 4 | $51.7 \pm 0.6$ | $9.6 \pm 0.4$ | $39.3 \pm 0.4$ | $26.0 \pm 1.2$ | $9.1 \pm 0.2$ |
| 5 | $51.9 \pm 0.6$ | $9.6 \pm 0.4$ | $39.1 \pm 0.4$ | $25.6 \pm 1.2$ | $9.0 \pm 0.2$ |

