# OpenReview forum: "Close the Gap: Lightweight Image Captioning via Retrieval Augmentation"
_ICLR.cc/2024/Conference — Submitted to ICLR 2024_

### Official Review · Reviewer_yd7b · 2023-10-31

**Soundness:** 1 poor
**Presentation:** 2 fair
**Contribution:** 1 poor
**Rating:** 3
**Confidence:** 4

**Summary:**

This paper focuses on image captioning with retrieval augmentation. Speicifically, the author proposed to retrieve a set of captions relevant to the input image via CLIP cross-modal retrieval. The retrieved captions are then sent into a pre-trained language model for summarization. The summarized results are regarded as the caption of the input image.

To improve the quality of CLIP retrieval, the author also proposed a light-weight projection layer that can be solved in closed-form to better align image and language modalities.

To augment the available caption set, the author also proposed an iterative self-improvement strategy by including captions generated by the model itself and improving the projection weight.

**Strengths:**

Training the projection weight $\textbf{W}$ is extremely low cost.

**Weaknesses:**

1. The main point of this paper is a $\textbf{lightweight}$ image captioning system. However, the proposed method is only lightweight during training in terms of training time. For inference, the model needs to encode the image, retrieve several captions, and then feed the retrieved captions into an LLM. The overall procedure may even be slower than a vanilla VL model for image captioning. For training, the proposed method did $\textbf{NOT}$ save trainable parameters compared to recent VL models like LLaVA, which only has a single fc layer to bridge the vision and language modalities. As the reviewer can tell from the paper, the "lightweight" part is only on the training speed and resources. This greatly reduced the contribution of this paper in the reviewer's opinion.

1. The performance of the proposed model is far below current SoTA models. Even though the proposed method saves training time and resources. The performance decrease is disproportional.

1. More details and analyses about the retrieval set should be discussed. For example, what retrieval set does the author use during training and testing/validation? What retrieval set does the author use for different datasets such as COCO and Flickr? Is the performance sensitive to the quality or distribution of the retrieval set?

1. There exists other retrieval augmented image captioning works that achieve reasonable performance such as REVEAL [1], RA-CM3 [2], HAAV [3], etc. Current approaches typically train the model jointly with the retrieval.

    [1] https://arxiv.org/abs/2212.05221

    [2] https://arxiv.org/abs/2211.12561

    [3] https://arxiv.org/abs/2305.16295

**Questions:**

Please see the weakness section, particularly about the details and analyses of the retrieval set.

---

> ### Author Response · Authors · 2023-11-23
>
> We address the mentioned weaknesses as follows:
>
> We **explicitly define lightweight** in the second paragraph of the introduction as the **number of trainable parameters.** The overall method is **NOT slower than other vanilla VL approaches.** We mention in Section 3.2 that inference is on-par with SmallCap (the most lightweight approach thus far). Furthermore, **LLaVa consists of a pre-training and a fine-tuning stage**, where the entire LM is trained during the latter. Even when considering only the pre-training stage of LLaVa, which only trains the linear mapping, **our method is more lightweight**, since the **embedding space of CLIP is 4 times smaller than the embedding space of LLama, which results in 4 times fewer trainable parameters. **
>
> We admit that we did not reach SOTA on any benchmark. However, our method is comparable with other lightweight captioning approaches, while being more efficient. Further, **we could slightly improve performance** by using a different prompting strategy, as highlighted in our revised version.
>
> Generally, the retrieval set (denoted datastore in the paper) contains the captions from the training set. This is the same during training and evaluation. Only in case when the self-improvement loop is applied, we add synthetic (i.e., self-generated) captions for the training images. We do not use additional data sources. To evaluate out-of-distribution performance we **will add results on NoCaps** in the revised version.
>
> Thank you for making us aware of additional relevant literature, we will add a discussion on these works in our revised manuscript.

---

### Official Review · Reviewer_ZWpM · 2023-11-08

**Soundness:** 2 fair
**Presentation:** 3 good
**Contribution:** 3 good
**Rating:** 3
**Confidence:** 3

**Summary:**

This work proposes a novel and cost-effective method to bridge the modality gap between pretrained language models (LMs) and pretrained vision-language models (VLMs) for enhanced image captioning. The modality gap refers to the misalignment between image and text representations in the shared embedding space, which can hinder the performance of image captioning models. The proposed method utilizes a linear mapping optimized via a least-squares solution to bridge this gap, enabling efficient computation on CPUs without requiring gradients. This approach allows for competitive performance on image captioning tasks compared to other lightweight captioning methods, particularly when reference-based metrics are employed.

**Strengths:**

The paper introduces a novel method of using a linear mapping to bridge the modality gap in image captioning, avoiding the need for costly end-to-end training. The proposed method requires only 1 million trainable parameters, which is significantly lower than other deep learning models, making it computationally efficient. The Iterative Self-improvement Loop Introduces a process of iteratively refining the mapping with synthetic captions generated by the model, which could lead to better performance metrics over time without additional human-labeled data. The method also achieves strong results on well-known benchmarks like MS-COCO and Flickr30k, indicating that the method performs well compared to existing lightweight captioning approaches. Overall, the paper provides a practical, efficient, and potentially more accessible solution to the problem of image captioning, with particular relevance to real-world applications where resources might be limited.

**Weaknesses:**

Generalizability Issue: The method is heavily relied on the referenced dataset which poses a critical limitation of the method. While the method is efficient, it may not generalize well to all types of images or domains, especially those significantly different from the reference dataset. If the testing dataset is very different from the reference set, it may cause catastrophic outcomes by error propagation and require much longer rounds of iteration. This would diminish the efficiency advantage. Further, if the reference dataset has limited topic/semantic coverage, then the potential hallucination issue would also happen as the LLM was only asked to summarize the referenced captions. Not matter how many iterations would be conducted, the hallucination may not be removed.

Limited Experiment Issue: Additionally, the performance is benchmarked on specific datasets. Since both Flickr and COCO are two very common datasets and share large similarity in terms of image distribution thus there might be a lack of evidence on how the model performs on out-of-distribution data or in more diverse real-world scenarios.

LLM Issue: The method also heavily relies on LLM, however the LLM never saw the image but only can summarize. The iterative refinement with synthetic captions could potentially introduce or amplify errors if the synthetic captions are not of high quality.

Computational Resources: While the model has fewer parameters, the computational cost and efficiency are not only about the number of parameters but also about the complexity of operations and the size of the input data. For every image, the method requires to pre-computation of all the reference image and text data, not to mention the following newly updated synthetic captions for every iteration.

Metric Sensitivity: The paper proposes a new metric for evaluation, but it’s not clear how sensitive the results are to the choice of metric and whether the proposed metric has been validated extensively.

Severely Limited Novelty/Details of the Caption: A good caption is not just a high-level summarization of the image but should also cover essential fine-grained details. The characteristics of this method determines it cannot include fine-grained details in the caption.

Novelty of the method: The method may lack novelty but is a practical improvement in real world scenarios.

**Questions:**

1. If the testing dataset is very different from the reference set, it may cause catastrophic outcomes by error propagation and require much longer rounds of iteration. This would diminish the efficiency advantage. How would you resolve this problem?

2. How would you guarantee the reference set necessarily has all the sufficient scenes against the testing set? If not, then certain critical scenes in testing set images can never be addressed properly. What is the criteria of selecting the reference set and testing set to ensure this method can work properly? If you cannot find a criteria, then this method has a fundamental issue.

3. Have you done experiment when the reference set and testing set are very different from each other?

Please see other questions in the  "Weakness" section.

**Details Of Ethics Concerns:**

No ethics concerns.

---

> ### Author Response · Authors · 2023-11-23
>
> We agree with the reviewer that our method is limited by the contents of the reference set (denoted datastore in the paper), that is the training set. However, it is **a general property of machine learning algorithms that their generalization abilities are restricted to the training distribution.** We do not claim that our method generalizes beyond the training distribution.
>
> We **will add results on NoCaps** for a linear mapping trained on the MS-COCO dataset to evaluate the out-of-distribution performance of our method.
>
> We agree that adding low-quality synthetic captions can amplify errors. This is the reason why **we threshold synthetic captions according to a certain metric**, which can be any of CIDEr-D, SPICE, B@4, ROUGE-L, etc. After additional experiments we found that **thresholding according to the CIDEr-D metric works the best** and usually leads to improvements across metrics.
>
> We believe that there is a misunderstanding regarding the self-improvement loop. The self-improvement loop is only part of the training algorithm and does not run during evaluation/inference. Further, our **reference set only contains texts which are pre-computed once.** During self-improvement, **synthetic captions are embedded once** and added to the reference set.  We will make this more clear in the paper.
>
> There must be a misunderstanding, since **we did not propose a new metric.**
>
> A new image **does not need to** be exactly covered by captions contained in the reference set. Several captions can contain objects contained in a new image and the **FLAN-T5 can interpolate between them** and find a suitable caption for a new image. We will add some qualitative examples on that in our revised version.
>
> **Questions:**
> - If a concept of a new image does not appear in the training captions then it will also not appear in the generated caption. However, **this problem is not specific to our method, but applies to any other image captioning pipeline.**
> - We believe there is a misunderstanding. Our method does **not rely on any additional data source.** It only retrieves from the training set, whereas other image captioning pipelines are trained end-to-end on the training data. As any other method **we rely on the assumption that the test distribution follows the training distribution.**
> - We will **add results on NoCaps** for a linear mapping trained on the MS-COCO dataset to evaluate the out-of-distribution performance of our method. Since our method uses the training captions as reference set, our method would not produce appropriate captions for test images that substantially differ from training images.

---

### Official Review · Reviewer_mspE · 2023-11-08

**Soundness:** 2 fair
**Presentation:** 2 fair
**Contribution:** 2 fair
**Rating:** 3
**Confidence:** 4

**Summary:**

This paper trains linear mapping (W) on top of the CLIP image encoder and text encoder to improve or refine the caption of a given dataset. During training, the mapping weight (W) uses the least-square error as a metric. During inference, the trained mapping will be used to rerank the top K similar captions from a pre-collected image-text pair pool, and then use a Large language model to refine the captions.

**Strengths:**

1. The topic of image captioning is important.
2. The proposed self-improvement loop is interesting.

**Weaknesses:**

The method presented is not one of image captioning; rather, it involves the enhancement of captions or the retrieval of text related to images. This is predicated on the assumption that there exists a preliminary set of fairly accurate captions for each image from which to draw. To better reflect this and to prevent any potential overstatement or confusion, a slight modification of the title is recommended.

The utility of the proposed method is constrained by a significant precondition: a repository of captions must be supplied to enable the retrieval process via mapping weights. Typically, we have only the image at our disposal without a pre-existing collection of initial or possible captions to draw from.

The principal innovation of this study is the introduction of a light linear mapping. However, this comes with two main limitations: (1) such a lightweight approach may be best suited to datasets that are small or of moderate size. The presumption here is that the learned weight
$W$ is sufficient to bridge the gap, which may not hold true for datasets larger than, for example, COCO—where a lightweight parameter might prove inadequate. (2) The drawback of a light parameterization is the necessity to maintain a substantial pool of captions to ensure coverage for any given image. Hence, the total memory required includes not just $W$ but also this extensive caption repository. Absent this, the method would be unable to perform image captioning autonomously.

The comparison in Table 1 appears somewhat skewed. The proposed method relies on the availability of a predetermined pool of captions for retrieval, in contrast to the baseline models which do not require such a resource.

**Questions:**

1 How to compare your method to the large multi-modal models, such as llava, instructBLIP, BLIP2, and MiniGpt-4? Any comparison on the image captions generated by those large multi-modal models?

---

> ### Author Response · Authors · 2023-11-23
>
> We address the mentioned weaknesses as follows:
>
> We believe there is a misunderstanding. After the **retrieval of related captions from the training set,** we use a LM (FLAN-T5) to generate a new caption for the input image. As long as the retrieved captions cover different aspects of the input image, the **LM can combine them to a new caption that fits the image.** We make this more clear in the revised version.
>
> Again, we believe that there is a misunderstanding. **Our method does not rely on any additional data source.** It only retrieves from the training set, whereas other image captioning pipelines are trained end-to-end on the training data. Thus, **training captions are generally available,** which is what we retrieve from during testing. Therefore, we **only require an input image** and no additional data. We try to clarify this in the revised version.
>
> In principle, we agree that there is no proof that a linear mapping is sufficient in all situations. However, there are prior works that **optimize a single linear layer on large-scale datasets [1,2].** This indicates that linear mappings are practically sufficient in similar settings. Moreover, these methods require **huge efforts to re-train a system of [1,2]** on new data, while for our method this is feasible within minutes. We agree that the datastore requires additional memory. However, **FAISS allows for compressing the datastore to reduce the memory footprint in case of large datasets.** We adapted the paragraph on the datastore in Section 5 accordingly.
>
> We believe the **comparison in Table 1 is adequate** since it compares our method to others that aim at **parameter efficiency, i.e. training as few parameters as possible.** All compared methods (including ours) only use the training set for training and evaluate on the test set. Further, **one of our baselines (SmallCap) uses retrieval** and stores the training data in its datastore (same as ReCap).
>
> **Caption quality compared to large-scale models:** We believe this is an unfair comparison, since **all of the mentioned methods train much more parameters on datasets that are orders of magnitude larger** than the ones we use. Further, they usually consist of **two-stage training** and use **language models of much larger size.**
>
> [1] Mini-GPT4: Enhancing vision-language understanding with advanced large language models, Zhu et al., arXiv:1909.11059, 2023
>
> [2] Visual Instruction Tuning, Liu et al., arXiv:2304.08485, 2023

---

### Official Review · Reviewer_RtHk · 2023-11-08

**Soundness:** 3 good
**Presentation:** 3 good
**Contribution:** 3 good
**Rating:** 5
**Confidence:** 5

**Summary:**

This paper targets the problem of lightweight captioning by proposing to optimize a linear mapping between the visual and text embedding space. They argue that this is crucial to close the modality gap. What differentiates this work from those tuning such embedding layer via a Cross-entropy loss is that they tune the linear mapping via simple least squares. The authors build a dataset of (image, text) pairs which can then be used to optimize for the linear mapping. One the linear mapping is obtained, they caption an image by first retrieving closest captions from a dataset and prompting these to an of-the-shelf LLM. They also propose a self-improvement phrase where the model is used to generate captions for the training set and then again used to refine the linear mapping. Authors claim similar performance to prior light-weight captioning methods by only tuning the linear layer and that too on a CPU in much less time. They also shows experiments on cross-dataset transfer and also ablation studies on which features to use for optimizing linear mapping. They also highlight an issue with clip-score metric.

Overall the idea seems relevant to the field. However, the method proposed in the paper lacks novelty and the experimental section is also weak.

**Strengths:**

- Addresses an important problem of designing light weight image captioning systems
- Proposes a training-free (at least no NN is trained) method to align visual and textual space. Argue that this closes the modality gap.
- Idea of using LLM to summarize the nearest NN captions for an image is interesting (although it can lack grounding esp. when the retrieved captions might be missing the key concept)
- Show comparable results on 2 datasets. Also, show that the self-improvement phrase results in improvement
- Highlights an issue with clip-score metric, where it seems to be assigning higher score to hallucinated example

**Weaknesses:**

- Overall the idea lacks novelty and depth. The key idea of doing retrieval augmented captioning is taken from prior work. Although the paper differs in how the training is being done (just doing least squares instead of training some layer), the performance is mostly below the previous works on most metrics (e.g. on Bleu small cap vs recap is 36 vs 29, numbers are similar on spice metric)
- Authors have shown experiments on cross-dataset transfer, why didn't they consider other datasets such wizwiz, MSR-VTT.
- Another issue that I see with this work is that the caption might not be grounded in the image. Based on Fig1, the final caption seems to be a summary of the retrieved captions which might not always be accurate. For example, what happens if the image contains a concept that is not a part of any caption
- I agree that the finding about clip-score is interesting but it was not adding value to the key idea in the paper

**Questions:**

Please see weakness section

---

> ### Author Response · Authors · 2023-11-23
>
> We address the mentioned weaknesses as follows:
>
> - We agree that retrieval augmentation has been done in prior works. However, these works require end-to-end training which we alleviate due to the linear mapping to bridge the modality gap. Further, **our self-improvement loop is a novel approach to augment the retrieval datastore with synthetic captions.**
>
>
> - We will add experiments on additional datasets, i.e. VizWiz and MSR-VTT in the future version.
>
>
> - If a concept of an image does not appear in the training captions then it will also not appear in the generated caption. However, **this problem is not specific to our method,** but applies to **any other image captioning pipeline.** The purpose of our linear mapping is to **ground images to  training captions.** In Table 4, it can be seen that without this grounding (ReCap⁻_Captions) there is a substantial drop in performance.
>
>
> - We believe that our analysis on CLIP-score does add value. The main reason we considered it is that we searched for an appropriate metric to **threshold our synthetic captions in the self-improvement loop.** We observed that **when optimizing for CLIP-score, our method started to hallucinate** which resulted in a decrease in all metrics except for CLIP-score itself.  We make this more clear in our revised version.

---

### Official Review · Reviewer_rPFf · 2023-11-09

**Soundness:** 2 fair
**Presentation:** 3 good
**Contribution:** 2 fair
**Rating:** 3
**Confidence:** 4

**Summary:**

In this research, the authors address the modality gap issue in pre trained vision-language models (VLMs) without the need for costly finetuning. They propose a cost-effective solution using a linear mapping optimized through a least-squares approach, achievable within minutes even on a CPU.  The method also includes an iterative refinement process using synthetic captions from the LM, allowing explicit optimization for image captioning metrics. The results demonstrate competitive performance on MS-COCO and Flickr30k datasets, especially in comparison to lightweight captioning approaches, and highlight the limitations of reference-free metrics such as CLIP-score.

**Strengths:**

The paper is well-written and easily comprehensible. However, there is room for improvement in conveying the analysis and intuition behind the discussed concepts.

While the paper addresses a crucial issue in the multimodal domain, the proposed solution's persuasiveness could be strengthened.

The author has presented qualitative and quantitative results across various datasets.

**Weaknesses:**

The author proposes bridging the modality gap through a cost-effective approach using a linear mapping optimized through a least-squares solution. However, there is a need for a more in-depth discussion on how this method differs from existing solutions in the realm of joint models.

The utilization of a "linear mapping optimized via a least-squares solution" is a fundamental constraint optimization technique. This may raise questions about the novelty of the method. To enhance the novelty of the method, the author should delve into a comparative analysis with existing joint model solutions found in the Visual Question Answering (VQA) literature and other multimodal joint representation studies. This discussion would shed light on the distinctive aspects and advancements introduced by the proposed approach.

Computational Complexity: The text mentions that certain computations, such as least-squares linear model fitting, can be done on a CPU within minutes. It would be valuable to provide a brief discussion or estimate of the computational complexity involved in each step, giving readers an idea of the method's efficiency.

Iterative Self-Improvement: The iterative self-improvement process is described, but the rationale behind choosing high-scoring captions for augmentation could be clarified. Explaining why high-scoring captions are selected and how this contributes to the refinement process would add depth to the method's justification.

Notation and Terminology: The use of symbols and notation is clear for the most part, but some terms could be defined or explained more explicitly. For instance, it would be helpful to explicitly state what "W" represents in the context of the linear model. Additionally, a brief glossary or notation table might aid readers in understanding the symbols used throughout the section. The hyperparameters "k" and "l" are introduced but not thoroughly explained. Providing more insight into the rationale behind choosing specific values for these hyperparameters or discussing their impact on the results would strengthen the method's transparency.

However, I feel that the paper misses one of the core aspects of machine learning practice: readability and reproducibility of results. What core mechanism of the proposed explanation method is not clear here. The author should provide an algorithm or pseudocode to reproduce the results, which this paper misses

**Questions:**

Please refer to the weakness section for this.

---

> ### Author Response · Authors · 2023-11-23
>
> We adress the mentioned weaknesses as follows:
>
> We **extend the related work section** to highlight the **differences to existing approaches to mitigate or avoid the modality gap.**
>
> To the best of our knowledge we are the **first to efficiently bridge the modality gap for image captioning** which, as opposed to existing approaches, alleviates end-to-end training of the captioning pipeline. We elaborate the **difference to other vision-language approaches** in the related work section. Further, **our self-improvement loop is novel** and provides a way to leverage synthetic captions for retrieval augmentation. Finally, our analysis on metrics provides a new insight that CLIP-score is not reliable for evaluation of image captioning.
>
> We added asymptotic complexity of the retrieval (O(n)) and the computation of the least squares solution via pseudoinverse (O(d³)), where d denotes the dimensionality of the CLIP space and n the number of embeddings stored in the datastore. We want to highlight that **neither of these components are a bottleneck of our method.** The **embeddings for the datastore are pre-computed once before training.** As mentioned in Section 3.2, inference time is approximately equal to the one of SmallCap, which, to the best of our knowledge, is the most lightweight approach that currently exists.
>
> We **threshold synthetic captions** in order to achieve a certain **trade-off between quality and diversity.** We ran additional experiments and found that thresholding according to the CIDEr-D metric works the best. Since CIDEr-D is based on n-grams, a high threshold would enforce captions as close as possible to reference captions, hence no additional information. Lowering the threshold results in more diverse synthetic captions that can be combinations of reference captions for different images.
>
> **W** is a **linear mapping from CLIP image space to the CLIP language space.** The hyperparameter **k** is the number of retrieved captions per image. We show in the appendix (Table 9 and Table 10) that it **slightly affects the downstream captioning performance.** The hyperparameter l is only used during the self-improvement cycle, to generate several **diverse captions.** In our revised version we avoid the need for this hyperparameter by using nucleus sampling.
>
> We agree that reproducibility is very important, which is why **we added our code in the supplementary material.** Further, to enhance understanding of our method, we **added algorithms for ReCap as well as the self-improvement loop.**

---

### Author Response · Authors · 2023-11-23

First, we want to thank all reviewers for reading our paper and providing feedback based on which we were able to substantially improve our paper.

We noticed that there seems to be a common misconception in how our proposed method works. The reasons for this misunderstanding reside in an ambiguous presentation from our side, which we believe is much more clear in the revised version. The misconception is that our method does not rely on any external data sources, but only retrieves from the regular training set. That is, we do not require access to any additional data.
To alleviate this confusion, we included the following changes (marked in red in our revision):

- added **algorithm for the captioning procedure**, clearly stating that the datastore used for inference only contains the training set
- added **algorithm for self-improvement loop**, clarifying thresholding of captions to ensure quality and clarifying that we only predict synthetic captions to augment the training set
- more **detailed discussion to existing VL approaches and modality gap** in the related work
- **improved results by a slight change in the prompting strategy**
- added **asymptotic complexity for retrieval and computation of mapping**

---

### Meta-Review · Area_Chair_6ATv · 2023-12-09

**Metareview:**

This paper was reviewed by five experts in the field and all reviewers recommend rejection. Reviewers liked the simplicity of the method but raised various concerns, mostly around the effectiveness of the proposed method. The authors' responses didn't sway the opinions of the reviewers.

The AC carefully read the paper, the reviews, and the authors' responses and agreed with the reviewers' assessments.

Considering the reviewers' concerns, we regret that the paper cannot be recommended for acceptance in its current form at this time. The authors are encouraged to consider the reviewers' comments when revising and resubmitting the paper.

**Justification For Why Not Higher Score:**

The reviewers unanimously recommended rejection and the AC didn't find strong reasons to overturn the reviewers' consensus.

**Justification For Why Not Lower Score:**

N/A

---

### Decision · Program_Chairs · 2024-01-16

Reject